# Variation in shortwave water vapour continuum and impact on clear-sky shortwave radiative feedback.

Kaah P. Menang[1,2], Stefan A. Buehler[2], Lukas Kluft[3], Robin J. Hogan[4] and Florian E. Roemer[2]

1. Department of Physics, University of Buea, Cameroon
2. Meteorological Institute, Department of Earth Sciences, Faculty of Mathematics, Informatics and Natural Sciences, University of Hamburg, Hamburg, Germany
3. Max Planck Institute for Meteorology, Hamburg, Germany
4. European Centre for Medium-Range Weather Forecasts, Reading, UK

*Correspondence to*: Kaah P. Menang (kaah.menang@ubuea.cm)

**Abstract.** This work assesses the impact of the current differences in the strength of the shortwave water vapour continuum on clear-sky calculations of shortwave radiative feedback. Three continuum models were used: the MT_CKD (Mlawer-Tobin-Clough-Kneizys-Davies; versions 2.5and 4.1.1) and CAVIAR_updated (Continuum Absorption at Visible and Infrared Wavelengths and its Atmospheric Relevance) models. Radiative transfer calculations were performed with the ECMWF radiation scheme ('ecRad'). The correlated $k$-distribution gas-optics tables required for ecRad computations were trained with each of these continuum models using the ECMWF correlated $k$-distribution software tool. The gas-optics tables trained with the different continuum models were tested individually in the shortwave. The atmosphere configuration was: fixed surface temperatures ($T_S$) between 270–330K, fixed relative humidity at 80%, a moist adiabatic lapse rate for the tropospheric temperature and an isothermal stratosphere with the tropopause temperature fixed at 175K. At $T_S$ =288K, it was found that the revisions of the MT_CKD model in the shortwave over the last decade and half have a modest effect (~0.3%) on the estimated shortwave feedback. Compared to MT_CKD_4.1.1, the stronger CAVIAR_updated model has a relatively greater impact; the shortwave feedback is ~0.006Wm$^{-2}$K$^{-1}$(~1.6%) more positive. Thus, for the clear-sky situation investigated, uncertainties due to the shortwave continuum is insignificant for present-day climate. The uncertainty in the shortwave feedback increases up to 0.008Wm$^{-2}$K$^{-1}$(~2.0%) between the MT_CKD models and 0.018Wm$^{-2}$K$^{-1}$(~4.6%) between CAVIAR_updated and MT_CKD_4.1.1 models at $T_S$≈300K. At $T_S$≈300K, a large portion of shortwave feedback originates at windows where CAVIAR_updated and MT_CKD_4.1.1 differ substantially leading to the larger feedback uncertainty.

## 1. Introduction

The concept of radiative-convective equilibrium (RCE), in which there is an energy balance between radiative cooling of the atmosphere and convective heating, is the simplest possible description of the climate system. Observations show that this idealisation of the climate system is a fairly accurate simplification of the tropical atmosphere (e.g., Popke et al., 2013; Kluft et al., 2019). Thus, different hierarchies of RCE atmospheric models have been used to study the Earth's tropical climate over the years (see, for example, Becker and Wing, 2020 and references therein). In particular, RCE models have in the last decade been used to investigate equilibrium climate sensitivity (ECS), radiative feedbacks, precipitation extremes and equilibrium climate as well the factors that influence them (e.g., Popke et al., 2013; Meraner et al., 2013; Reed et al., 2015; Dacie et al., 2019; Kluft et al., 2019; Becker and Wing, 2020; Wing et al., 2020).

Despite having the fewest number of interactive processes within the RCE model hierarchy, the one-dimensional (1D) RCE model is fundamental in obtaining the first estimates of radiative feedbacks and ECS, which are important for understanding climate and climate change of the tropics (e.g., Ramanathan and Coakley, 1978). For example, Manabe and Wetherald (1967) used a 1D RCE model to robustly estimate the ECS and water vapour feedback that have stood the test of time. However, there have been significant disagreements over the values of these (and other) quantities calculated from 1D RCE models (e.g., Schlesinger, 1986; Kluft et al., 2019). Factors that may lead to uncertainties in ID RCE calculations of these climate parameters include: vertical relative humidity (RH) distribution, concentration of other atmospheric gases, clouds and radiative transfer calculations. Recently, Bourdin et al. (2021) and Kluft et al. (2019) have shown that differences in the vertical distribution of RH have a significant impact on 1D RCE clear-sky calculations of radiative feedbacks and ECS. In another study, Dacie et al. (2019) concludes that 1D RCE clear-sky calculations of the tropical tropopause layer and surface climate are sensitive to $CO_2$ concentration and ozone profile. Kluft (2020) showed that the presence of clouds has a significant impact on 1D RCE calculations of radiative feedbacks and forcings. Kluft et al. (2019) speculated that the differences in the treatment of radiative transfer may be an additional reason for some discrepancies in estimated climate quantities, such as ECS, by different 1D RCE studies. But they did not specify the aspects of radiative transfer that could contribute to these uncertainties.

The atmospheric absorption by water vapour is an essential aspect of radiative transfer calculations in 1D RCE (and other climate) models (e.g., Kratz, 2008). Unlike the water vapour spectral-line absorption which is well-understood, absorption by the water vapour continuum is still uncertain, with larger uncertainties in atmospheric windows at shorter wavelengths (e.g., Shine et al., 2016). Uncertainties in water vapour continuum absorption have contributed to discrepancies in the estimation of the shortwave absorption from climate models, with an increase in continuum absorption in recent global climate models leading to an increase in shortwave absorption and better agreement with observations (e.g., Wild, 2020 and references therein). Using the Community Earth System Model, Kim et al. (2022) showed that an increase in shortwave absorption by water vapour could lead to a reduction in the global-mean rainfall. However, it should be noted that Kim et al. (2022) did not directly associate this increase in water vapour absorption to the continuum.

Most radiative transfer codes used in climate models parameterise water vapour continuum by the semi-empirical MT_CKD (Mlawer-Tobin-Clough-Kneizys-Davies) model (Mlawer et al., 2012, 2023). Different versions of the MT_CKD model are used in climate models even though the strengths of the water vapour continuum in these versions are significantly different, especially in shortwave spectral regions. There are also significant disagreements between the strengths of the MT_CKD model and other currently available water vapour continuum models (e.g., Elsey et al., 2020 and associated references). In fact, some recent studies have pointed out that the MT_CKD model may be underestimating the strength of the water vapour continuum in some near-infrared atmospheric windows (e.g., Elsey et al., 2020). An increase in water vapour continuum absorption is expected to have more impact on the tropical atmosphere, since it has a higher water vapour content than the other atmospheres.

A few studies have investigated the effect of longwave water vapour continuum absorption on longwave radiative feedback (e.g., Koll et al., 2023; Roemer et al., 2024). Koll et al. (2023) investigated the effect of the water vapour continuum feedback on the net clear-sky longwave feedback. Using a line-by-line radiative transfer model that includes the MT_CKD continuum model, version 3.2, Koll et al. (2023) reported that the impact of the water vapour continuum feedback (which is positive and hence has a destabilising effect) on the net longwave feedback depends on RH. Compared to calculations at high RH, Koll et al. (2023) found out that a reduction in the continuum feedback at low RH leads to a significant increase in the net longwave feedback. Roemer et al. (2024) studied how the uncertainty in characterising water vapour continuum absorption affects clear-sky longwave feedback. Using

a line-by-line radiative transfer model and a single-column atmospheric model, Roemer et al. (2024) concluded that a 10 % uncertainty in MT_CKD continuum model, version 4.0 leads to a modest effect (~0.1 %) on longwave feedback at a surface temperature of 288 K. At surface temperatures greater than 300 K, this effect increases to about 7%. However, the impact of water vapour continuum absorption in the shortwave spectral region on shortwave radiative feedback has been given relatively little attention. In the last decade, Radel et al. (2015) is seemingly the only study that has looked at the impact of the water vapour continuum on shortwave feedback. Using a broadband radiation scheme, Radel et al. (2015) concluded that differences in representing water vapour continuum absorption at near-infrared and visible wavelengths by two (older and no longer used) models could lead to a 7% difference in estimated clear-sky shortwave feedback. Thus, the impact of recent changes of water vapour continuum models at these wavelengths on shortwave feedback has been neglected. This is clearly an oversight since the contribution of shortwave water vapour continuum absorption to radiative feedback in a warming climate is non-negligible (e.g., Jeevanjee, 2023). This paper presents an investigation on the impact of the uncertainty in the representation of the shortwave water vapour continuum on the clear-sky calculations of shortwave radiative feedback using a 1D RCE model. The radiative transfer calculations of this RCE model were performed using the fast and accurate European Centre for Medium-Range Weather Forecasts (ECMWF) radiation scheme (Hogan and Bozzo, 2018). The correlated $k$-distribution gas-optics tables for the radiative transfer calculations were each trained with the water vapour continuum models selected for this work. These gas-optics tables were generated using the ECMWF correlated $k$-distribution ('ecCKD') software tool developed recently by Hogan and Matricardi (2022). The rest of this paper is structed as follows: Section 2 will focus on the data and methods. In Section 3, the results from this work will be presented. Section 4 summarises and concludes.

## 2. Data and methodology

### 2.1. Water vapour continuum formulation

As stated in Section 1, different versions of the MT_CKD water vapour continuum model exist. The uncertainty in characterising water vapour continuum absorption has led to updates of this continuum model over the years. Here, we briefly discuss the significant updates in the shortwave spectral region. The MT_CKD model was originally derived by adjusting the water vapour monomer line shape with observed continuum absorption coefficients in the mid- and

far-infrared spectral regions using a spectrally varying function called the 'χ function' (Mlawer et al., 2012). The modified line shape was then applied to all water vapour lines out to near-infrared and visible wavelengths, giving a water vapour continuum in spectral regions where no measurements were available. However, in the 2500 cm$^{-1}$ window, scaling factors were applied to the MT_CKD model, version 2.5 (MT_CKD_2.5; Mlawer et al., 2012) to achieve agreement with some observations in this spectral region (see Mlawer et al., 2012, for details). A good number of measurements showed that the MT_CKD_2.5 model was underestimating the self-continuum absorption in the near-infrared (e.g., Ptashnik et al., 2011; Campargue et al., 2016; Reichert and Sussmann, 2016). Optical-feedback-cavity enhanced absorption spectroscopic (OFCEAS) and cavity ring-down spectroscopic (CRDS) laboratory measurements, considered to be highly accurate, of the self-continuum absorption coefficients at near-infrared windows (Lechevallier et al., 2018 and associated references) were used to adjust the MT_CKD model, version 3.2 (MT_CKD_3.2; Mlawer et al., 2012) in the near-infrared. The MT_CKD_3.2 model was a major revision of the MT_CKD model and is generally stronger than the MT_CKD_2.5 model at near-infrared windows (see, for example, Figure 7 of Elsey et al., 2020). That notwithstanding, further CRDS measurements by Vasilchenko et al. (2019) suggested that MT_CKD_3.2 may be overestimating the self-continuum at near-infrared windows close to 5130 cm$^{-1}$ and 5700 cm$^{-1}$. Version 4.1.1 of the MT_CKD model (MT_CKD_4.1.1; Mlawer et al., 2023) is the most recent version of the MT_CKD model at the time this work was done. In the near-infrared, the strength of the self-continuum in MT_CKD_4.1.1 is equal to that of MT_CKD_3.2 (see Figure 2d, Mlawer et al., 2023). However, the temperature dependence of the self-continuum was reformulated in MT_CKD_4.1.1.

Several measurements concluded that the MT_CKD_2.5 model was also underestimating the foreign-continuum at near-infrared windows (e.g., Ptashnik et al., 2012; Mondelain et al., 2015; Reichert and Sussmann, 2016). The MT_CKD_3.2 near-infrared foreign-continuum was adjusted using the CRDS measurements of the foreign-continuum absorption coefficients at near-infrared windows by Mondelain et al. (2015). However, more CRDS and OFCEAS measurements showed that MT_CKD_3.2 may still be underestimating the foreign-continuum absorption at near-infrared windows (Vasilchenko et al. 2019; Fleurbaey et al., 2022). The MT_CKD_4.1.1 foreign-continuum is also equal to the MT_CKD_3.2 foreign-continuum at near-infrared wavelengths (Mlawer et al., 2023, Figure 2d).

The self- and foreign-continua coefficients of the MT_CKD_2.5 model were modified with those from extrapolated higher temperature Fourier transform spectrometer laboratory

measurements at near-infrared windows by Ptashnik et al., (2011, 2012) to produce the CAVIAR (Continuum Absorption at Visible and Infrared Wavelengths and its Atmospheric Relevance) water vapour continuum model. The self-continuum of this model was updated by Jon Elsey and Keith Shine in 2020 using a combination of continuum coefficients from MT_CKD_3.2 and laboratory measurements of Paynter et al. (2009) and Ptashnik et al. (2011,

2019) (Keith Shine, 2023, personal communication). This updated version of CAVIAR model (referred to as 'CAVIAR_updated' model henceforth) was used in this study.

We remark here that even though both the water vapour self- and foreign- continua absorb solar radiation, the self-continuum is more important than the foreign-continuum for this work. This is because under clear-sky conditions, shortwave absorption is dominated by

180 the water vapour self-continuum (e.g., Paynter and Ramaswamy, 2011). However, any water vapour continuum model used in this work is made up of both the self and foreign- continua.

The CAVIAR_updated continuum model and two versions of the MT_CKD continuum model (MT_CKD_4.1.1 and MT_CKD_2.5) were selected for this study. The MT_CKD_2.5 model, released in 2010, is arguably the most widely used version of the MT_CKD continuum

model and is still used in some climate models today. For example, it is used by the UK Met Office Unified Global Atmosphere 7.0/7.1 model (Walters et al., 2019). Although the MT_CKD_3.2 model (released in 2017) includes a major revision of the self-continuum compared to MT_CKD_2.5, it is similar to the MT_CKD_4.1.1 model (released in 2023) in the near-infrared region as discussed above. Thus, only the most recent MT_CKD_4.1.1 model

was selected for this work in addition to version 2.5. After this work was completed, a revision of the near-infrared water vapour foreign-continuum was made to version 4.3 of the MT_CKD model (see, https://github.com/AER-RC/MT_CKD_H2O/wiki/What's-New), that would have some impact on the absorption of solar radiation. But this impact is not expected to significantly affect the results from this work (see Section 3) since the foreign-continuum is relatively

weaker than the self-continuum.

Note that the strengths of other publicly available water vapour models, such as the BPS-MTCKD model (version 2.0; Paynter and Ramaswamy, 2014), fall within the range of the models selected here and thus there was no added value including them in this study. Figure 1 (a) shows the absorption coefficients of the continuum models selected for this work from

200 about 0 – 20,000 cm$^{-1}$ in the 920 m thick atmospheric layer between ~960 hPa and ~860 hPa for the "median profile" of the Correlated K-Distribution Model Intercomparison Project datasets (CKDMIP; Hogan and Matricardi, 2020) while Figure 1(b) shows the absorption coefficient ratio of the other two continuum models with that of MT_CKD_4.1.1 model. The

CAVIAR_updated model is generally stronger than the MT_CKD models (Figure 1(a)) in most near-infrared atmospheric windows and some bands. Figure 1(b) shows that the CAVIAR_updated model is much stronger than the MT_CKD_4.1.1 model in near-infrared windows, where it is up to a factor of 7 stronger at the 6000 cm$^{-1}$ (1.6 $\mu$m) window. But at the edges of the windows around 4000 cm$^{-1}$, 6000 cm$^{-1}$ and 8000 cm$^{-1}$, the MT_CKD_4.1.1 model is slightly stronger than the CAVIAR_updated model. The MT_CKD_2.5 model is also stronger than the MT_CKD_4.1.1 in the 2500 cm$^{-1}$ (5 $\mu$m) window, but weaker in most regions of the shortwave.

Figure 1(c) shows the terrestrial spectrum from 0 – 2000 cm$^{-1}$ and solar spectrum from 2000 – 20000 cm$^{-1}$ for a tropical atmosphere. Comparing these irradiances with the absorption coefficient ratios in Figure 1(b), it can be clearly seen that, in the shortwave, the continuum models differ most at near-infrared windows from about 2500 – 10000 cm$^{-1}$.

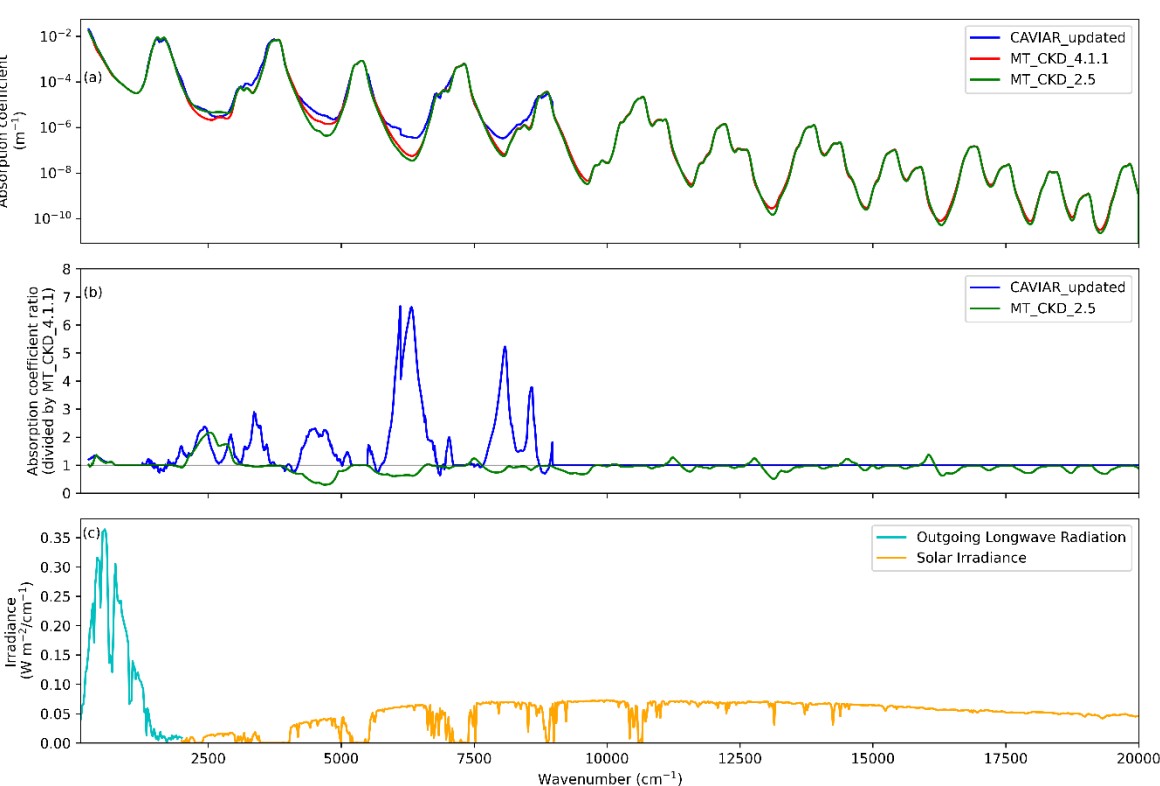

**Figure 1. (a) Atmospheric absorption coefficients of CAVIAR (blue), MT_CKD_4.1.1 (red), and MT_CKD_2.5 (green) continuum models from 0 – 20,000 cm$^{-1}$. (b) Absorption coefficient ratios of the other continuum models with that of the MT_CKD_4.1.1 model. These absorption coefficients are for the 920 m thick atmospheric layer between ~960 hPa and ~860 hPa with temperatures about 289 K and 286 K and water vapour mole fraction about 0.0137 and 0.0101 and for the median profile of the CKDMIP datasets. The**

**absorption coefficients were calculated by simply dividing layer optical depths of the**
**CKDMIP median profile by the layer thickness of 920 m calculated with the Hypsometric**
**equation. (c) Spectral outgoing longwave radiation from the top-of-the-atmosphere**
**(cyan) and spectral solar irradiance at ~960 hPa (orange). These irradiances were**
**calculated for a tropical atmosphere using the Atmospheric Radiative Transfer Simulator**
**(ARTS; Buehler et al., 2018, 2025).**

## 2.2. Radiation scheme

The Python code konrad (Dacie et al., 2019; Kluft et al., 2019) was used to construct the moist adiabatic atmosphere for the different surface temperatures (see Section 2.3) and to call the radiation scheme.

The default code for radiative transfer calculations in konrad is RRTM for GCM applications (RRTMG; Mlawer et al., 1997), a radiation scheme that is widely used in many dynamical models such as the Community Earth System Model (Kay et al., 2014) and the ECHAM6 model of the Max Planck Institute for Meteorology, Hamburg, Germany (Stevens et al., 2013).

However, for this study, the offline version of the European Centre for Medium-Range Weather Forecasts (ECMWF) radiation scheme ('ecRad'; Hogan and Bozzo, 2018) was used in konrad through Python subprocesses. ecRad is an efficient, flexible, and fast radiation scheme that is currently used in ECMWF Integrated Forecast System (IFS) and other models such as ICON (Icosahedral Nonhydrostatic) of the German Weather Service. Five solvers, including a 'cloudless' solver, are currently available in ecRad. The flexibility of this radiation scheme, which is based on its modular structure, enables it to be adapted for different uses. This flexibility was exploited to use any of the generated gas-optics models during each konrad run. Also, since the focus of this study is on clear-sky conditions, radiative transfer calculations were performed using the 'cloudless' solver of ecRad.

## 2.3. Model configuration and calculations

Unless otherwise specified, all the calculations carried out in this study use the following configuration of konrad.

The atmosphere was constructed on 512 levels of temperature and water-vapour volume mixing ratio between 1000 and 0.01 hPa. As recommended by the Radiative Convective

Equilibrium Model Intercomparison Project (RCEMIP; Wing et al., 2018), the reduced total solar irradiance was set to 551.58 W m$^{-2}$ and the zenith angle to 42.05°, resulting in an insolation of 409.6 W m$^{-2}$, which is the annual mean value in the tropics. The surface in the model has a fixed surface temperature with prescribed values from 270 to 330 K (in 1 K increments) and an albedo of 0.2. Atmospheric temperature is set to follow a moist adiabat in the troposphere with the tropopause temperature fixed at 175 K. This moist adiabatic temperature profile is consistent with the assumption of RCE (e.g. Jeevanjee, 2023). Since the stratosphere is more in radiative equilibrium rather than RCE, it was represented as an isothermal layer with this fixed tropopause temperature. This restriction also eliminates any stratospheric feedbacks. This is the same setup as in, for example, Kluft et al. (2021), Jeevanjee (2023) and Roemer et al. (2024).

The relative humidity (RH) is set at a constant value of 80% throughout the troposphere up to the cold-point tropopause. For the tropics, this value is higher than the observed average value of about 40% from the mid to upper troposphere (e.g., Bourdin et al., 2021), but it is chosen to ensure that the amount of humidity in the upper levels of the troposphere is adequate for the interaction of lapse-rate and water-vapour feedbacks (Kluft et al., 2021). In the stratosphere, the specific humidity is kept constant at the value obtained at the cold-point tropopause. The concentrations of the other trace gases used are those specified by Kluft et al. (2019), including a $CO_2$ concentration of 348 ppmv and the ozone concentration profile defined according to RCEMIP guidelines.

The shortwave climate feedback parameter ($\lambda_{SW}$) is calculated using the fixed-temperature method (Kluft et al., 2021) at a constant $CO_2$ concentration of 348 ppmv and surface temperature, $T_S$, from:

$$\lambda_{SW} = \frac{\Delta R\left(T_s + \Delta T\right) - \Delta R\left(T_s - \Delta T\right)}{2\Delta T}, \tag{1}$$

where $\Delta R$ is the net shortwave radiation at the top of the atmosphere and $\Delta T = 1$ K. Kluft et al. (2021) have justified the use of this fixed-temperature method by proving that results from it are in a very good agreement with those from the more frequently used linear regression method of Gregory et al. (2004) and have the advantage of being numerically more stable. For each $T_S$, we adjust the tropospheric temperature ($T$) and water vapour mixing ratio ($q$) to the moist adiabat and calculate the clear-sky shortwave radiation. We then calculated $\lambda_{SW}$ from Equation (1).

## 3. Results and discussion

For the atmospheric configuration described in Section 2.3, experiments were carried out in which gas-optics tables trained with the MT_CKD_2.5 and CAVIAR_updated continuum models were each used for the shortwave radiation. The effect of the shortwave water vapour continuum uncertainty on the estimation of radiative feedback was obtained by comparing results from experiments in which MT_CKD_2.5 and CAVIAR_updated models were used with those from an experiment in which the MT_CKD_4.1.1 trained gas-optics table was used.

### 3.1 Shortwave radiative feedback for present-day surface temperature

At $T_S$ = 288 K, the clear-sky shortwave climate feedback parameters ($\lambda_{SW}$) calculated are; 0.364 W m$^{-2}$ K$^{-1}$, and 0.371 W m$^{-2}$ K$^{-1}$ for the MT_CKD_2.5 and CAVIAR_updated continuum models respectively. For the reference experiment with the MT_CKD_4.1.1 trained gas-optics table, $\lambda_{SW}$ = 0.365 W m$^{-2}$ K$^{-1}$. If needed, the total radiative feedback parameter for each continuum model from this study can be readily obtained by using the additive property of feedbacks; total feedback, $\lambda_{tot} = \lambda_{LW} + \lambda_{SW}$, where the clear-sky longwave feedback $\lambda_{LW}$ can be obtained from similar 1D RCE studies for present-day average surface temperatures, such as, those by Xu and Koll (2024), Koll et al. (2023) and Kluft et al. (2021).

Under clear-sky conditions, the shortwave feedback is due to the absorption of solar radiation by water vapour. Shortwave absorption by water vapour increases in a warmer world, because the amount of water vapour in the atmosphere increases (scaling with the Clausius-Clapeyron relation for a fixed RH). This increased absorption of solar radiation thus gives an extra positive feedback in climate change. A stronger shortwave water vapour continuum means a further increase in absorbed solar radiation in a warming world leading to a stronger positive shortwave feedback. Thus, the shortwave radiative feedback tends to be more positive with increasing strength of the shortwave water vapour continuum as the values of $\lambda_{SW}$ given above show.

The values of $\lambda_{SW}$ obtained here show that compared to MT_CKD_4.1.1, differences in the strength of the shortwave continuum in the MT_CKD model make a modest contribution to the estimated shortwave radiative feedback. The shortwave feedback using the weaker MT_CKD_2.5 model is only about 0.001 W m$^{-2}$ K$^{-1}$ (~0.3 %) less positive than that with MT_CKD_4.1.1. However, relative to MT_CKD_4.1.1, the stronger CAVIAR_updated shortwave continuum model increases the shortwave feedback by about 0.006 W m$^{-2}$ K$^{-1}$ (~1.6 %). This effect is smaller than that obtained by Radel et al. (2015). This is because, in the

shortwave, Radel et al. (2015) used the Clough–Kneizys–Davies (CKD) model which is weaker than MT_CKD_4.1.1 and a version of the CAVIAR model that is stronger than that used here. Also, the shortwave irradiances in Radel et al. (2015) are globally averaged.

Thus, if the shortwave water vapour continuum absorption is as strong as suggested by the CAVIAR_updated model, then there may be a slight underestimation of clear-sky shortwave radiative feedback from 1D RCE models with the MT_CKD continuum model for present-day surface temperature of 288 K.

**3.2. Temperature-dependence of the shortwave radiative feedback**

As discussed in Section 3.1, the shortwave radiative feedback depends quite strongly on the surface temperature $T_S$ because there is more moisture in a warmer atmosphere. In this section, the impact of changing $T_S$ on $\lambda_{SW}$ will be presented. As a function of $T_S$, Figure 2 (a) shows the variation of $\lambda_{SW}$ for all three continuum models considered in this study, while Figure 2 (b and

335 c) show respectively, the absolute and percentage continuum induced difference in $\lambda_{SW}$ (with respect to calculations using MT_CKD_4.1.1). Figure 2(a) shows that $\lambda_{SW}$ increases with $T_S$ for all continuum models (from slightly less than 0.300 W m$^{-2}$ K$^{-1}$ at 270 K to slightly above 0.750 W m$^{-2}$ K$^{-1}$ at 330 K). Since the atmospheric water vapour content increases with temperature, at higher temperatures the increased atmospheric moisture reduces the upwelling

shortwave radiation leading to an increase in the net shortwave radiation at the top of the atmosphere and hence $\lambda_{SW}$. This figure shows that the $\lambda_{SW}$ due to the use of CAVIAR_updated model is not always greater than the $\lambda_{SW}$ estimated using the other models at all surface temperatures as hinted by calculations at $T_S = 288$ K.

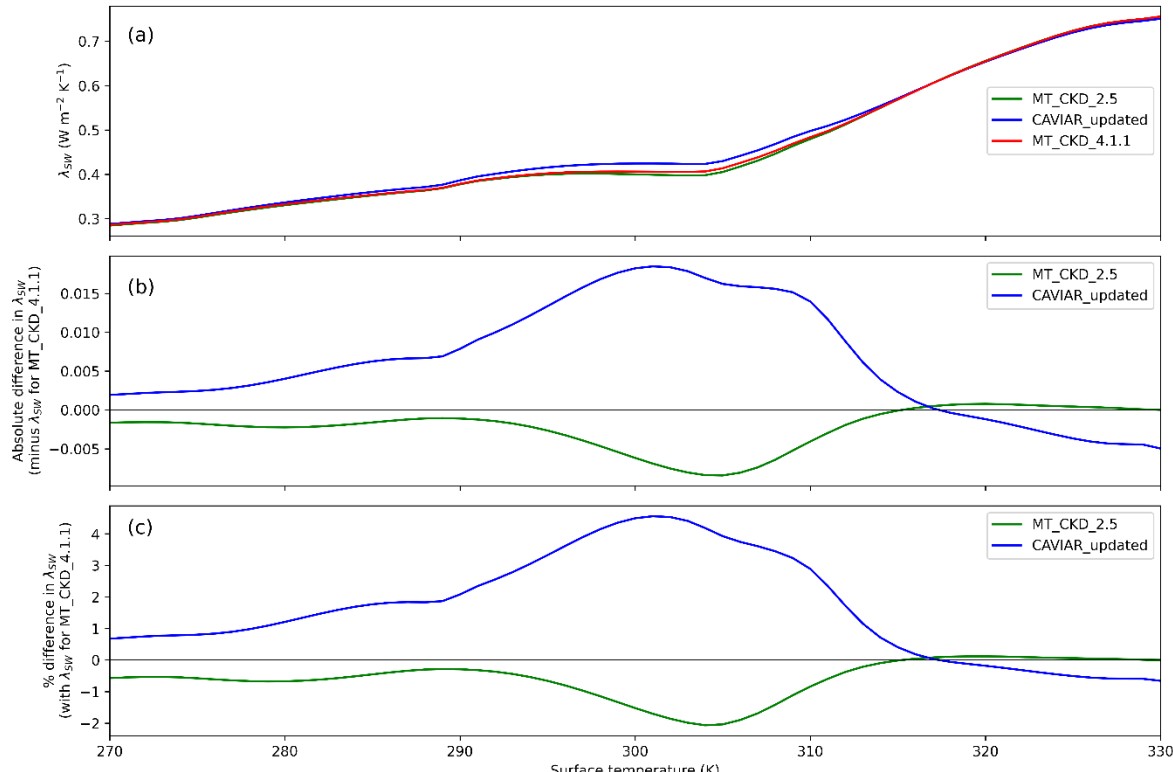

**Figure 2. As a function of surface temperature ($T_S$), (a) variation of clear-sky shortwave climate feedback parameter ($\lambda_{SW}$) estimated using the MT_CKD_2.5 (green), CAVIAR_updated (blue) and the chosen reference MT_CKD_4.1.1 (red) continuum models; (b) absolute difference in $\lambda_{SW}$ (with respect to values calculated using MT_CKD_4.1.1 model); (c) percent differences in $\lambda_{SW}$ (with respect to calculations using MT_CKD_4.1.1).**

Figure 2(b and c) show that the relative difference in $\lambda_{SW}$ that is induced by the continuum uncertainty also depends relatively strongly on the surface temperature. Compared to shortwave feedback calculated with the MT_CKD_4.1.1 model, the shortwave feedback with the MT_CKD_2.5 model is up to ~0.008 W m$^{-2}$ K$^{-1}$ (~2.0 %) less at a surface temperature of ~305 K. From $T_S$ of about 305 – 315 K, the magnitude of the relative difference in estimated shortwave feedbacks due to differences in the strength between these two continuum models is less than this value. For $T_S$ greater than ~318 K, the shortwave feedbacks with both models are equal and the uncertainty in $\lambda_{SW}$ due to these continuum models is zero.

The estimated shortwave feedback with the CAVIAR_updated model is up to ~0.018 W m$^{-2}$ K$^{-1}$ (~4.6 %) more positive than that with the MT_CKD_4.1.1 model at a surface temperature of ~301 K as Figure 2 (b and c) show. From $T_S$ of about 301 – 316 K, this relative difference decreases with surface temperature. Beyond $T_S$ of about 316 K, the feedback with

CAVIAR model is less positive than that with MT_CKD_4.1.1 model and increases negatively with temperature (but only up to about 0.005 W m$^{-2}$ K$^{-1}$ or ~0.7 % less). Beyond ~316 K, the shortwave feedback from CAVIAR_updated is less due to the decreased absorption in CAVIAR_updated compared to MT_CKD_4.1.1 at the edges of the windows above 4000 cm$^{-1}$, which are the spectral region that probably contributes most to $\lambda_{SW}$ at higher $T_S$, as discussed below.

The way the continuum induced uncertainty in $\lambda_{SW}$ changes with surface temperature (Figure 2(b and c)) can presumably be explained by spectral variations in both the water vapour spectroscopy and in the differences between the CAVIAR_updated and MT_CKD models. Shortwave water vapour absorption in windows between absorption bands generally tends to decrease with increasing wavenumber (Figure 1(a)). At the same time, the largest discrepancies between CAVIAR_updated and MT_CKD occur at intermediate wavenumbers within the near-infrared, that is, in the atmospheric windows around 4000 cm$^{-1}$, 6000 cm$^{-1}$, and 8000 cm$^{-1}$, respectively. This is relevant because, broadly speaking, $\lambda_{SW}$ presumably originates at wavenumbers at which the transmissivity of the atmosphere changes most rapidly, that is, at column-integrated opacities of order unity ($\tau_{col} \approx 1$). At low $T_S$ – and thus low water vapour concentration – $\tau_{col} \approx 1$ primarily occurs within the strong absorption bands below 4000 cm$^{-1}$, where CAVIAR_updated and MT_CKD are not very different and thus the difference in $\lambda_{SW}$ is small. As $T_S$ – and thus water vapour concentration – increases, $\tau_{col} \approx 1$ more frequently occurs in the above mentioned windows where CAVIAR_updated and MT_CKD differ substantially. Consequently, a large portion of $\lambda_{SW}$ originates in these windows at $T_S$ around 300 K, contributing to the substantial differences in estimated $\lambda_{SW}$ there. At even higher $T_S$, $\tau_{col} \approx 1$ occurs at even higher wavenumbers, where the differences between CAVIAR and MT_CKD are small, and thus the differences in $\lambda_{SW}$ is also small. We remark that CAVIAR_updated is very close to MT_CKD at high wavenumbers because it was constructed to agree with MT_CKD above ~8000 cm$^{-1}$, as discussed in Section 2.1. A few water vapour continuum measurements have been done at wavenumbers higher than 8000 cm$^{-1}$ (e.g., Campargue et al., 2016; Fulghum and Tilleman, 1991). Thus, the modified water vapour line shape used to derive the MT_CKD (and hence CAVIAR_updated) continuum absorption coefficients at these wavenumbers has been constrained with very limited measurements in this region. The limited understanding of water vapour continuum absorption at these high wavenumbers may also be responsible for its absorption having little or no impact on estimated $\lambda_{SW}$ at $T_S$ above about 316 K (Figure 2(b and c)).

The shortwave feedbacks calculated here at temperatures above ~296 K are subject to an uncertainty. The temperature dependence of the water vapour self-continuum in ecRad is implemented by applying the parametric fits of the continuum models in which the temperature is defined by specifying the values at 260 and 296 K (e.g., Mlawer et al., 2023). This increases the error in the self-continuum and hence estimated shortwave feedbacks for surface temperatures above ~296 K.

Roemer et al. (2025) have recently developed a conceptual model of the temperature dependence of clear-sky shortwave water vapour feedback and shown that it agrees well with that from line-by-line radiative transfer calculations using ARTS, which incorporates MT_CKD (version 4.0, that is equal to MT_CKD_4.1.1 in the shortwave) water vapour continuum model. Our results based on ecRad are consistent with those of Roemer et al. (2025) up to ~310 K but deviate above. We hypothesize that this discrepancy is due to the fact that the correlated-$k$ model uses a fixed (and relatively small) set of wavenumbers for computing the shortwave radiation. Under global warming, successively different parts of the solar spectrum become relevant for the feedback (but always the wavenumbers that transition from optically thin to optically thick as discussed above). Thus, the correlated-$k$ model runs out of a good number of wavenumbers at larger temperatures and diverges from a full radiative transfer model. This discrepancy in temperature dependence of $\lambda_{SW}$ will be investigated further in another study.

## 4. Summary and conclusions

Radiative transfer calculations for a moist adiabatic troposphere at different surface temperatures have been used to study the impact of the differences in the strength of the water vapour continuum in the shortwave spectral region on shortwave radiative feedback. Two versions of the MT_CKD (versions 2.5 and 4.1.1) and the CAVIAR_updated water vapour continuum models were selected for this work. This effect was studied through radiative transfer calculations using shortwave correlated-$k$ gas-optics tables trained with different continuum models. The ECMWF's fast and accurate radiation scheme was used for the radiative transfer calculations.

For present-day average surface temperature of 288 K, an increase in the strength of the shortwave water vapour continuum leads to a more positive shortwave feedback. At this temperature, the discrepancies in the strength of the shortwave water vapour continuum leads to very small uncertainties in the estimated shortwave radiative feedback between the two

MT_CKD models. Differences of only up to about 0.001 W m$^{-2}$ K$^{-1}$ (~0.3 %) in estimated shortwave feedback are obtained between these models. Compared to calculations with the MT_CKD_4.1.1 model, the shortwave feedback due to the use of the stronger CAVIAR_updated continuum is about 0.006 W m$^{-2}$ K$^{-1}$ (~1.6 %) more positive.

The estimated shortwave radiative feedback depends on surface temperature. The relative difference in shortwave feedbacks due to changes in the MT_CKD continuum models increases with surface temperature to a maximum of ~0.008 W m$^{-2}$ K$^{-1}$ (~2.0 %) at $T_S \approx 305$ K. When the CAVIAR_updated continuum model is used, the shortwave feedback uncertainty increases with surface temperature to a maximum of ~0.018 W m$^{-2}$ K$^{-1}$ (~4.6 %) at $T_S \approx 300$ K, when compared with that estimated using MT_CKD_4.1.1. The continuum induced difference in $\lambda_{SW}$ is small at low $T_S$ (close to 270 K) and at high $T_S$ (above ~300 K) because of the spectral variations in both the water vapour spectroscopy and in the differences between the CAVIAR_updated and MT_CKD models.

Therefore, for the clear-sky situation investigated, uncertainties in the shortwave water vapour continuum do not contribute significant uncertainty to climate sensitivity for present day temperatures. But for a much warmer world, uncertainty due to the shortwave continuum will have a relatively significant impact.

The results from this study show that the revisions of the MT_CKD water vapour continuum model in the shortwave over the past decade and a half have only a small effect on the clear-sky shortwave radiative feedback computed from a 1D RCE model for present-day average surface temperature. Compared to the MT_CKD_4.1.1 model, the stronger CAVIAR_updated water vapour continuum model has a relatively greater impact on the shortwave radiative feedback. For higher surface temperatures, the impact of uncertainties of water vapour continuum on the estimation of shortwave feedbacks is higher. Thus, using the MT_CKD model in RCE models may lead to an underestimation of the shortwave feedback, if it is underestimating shortwave water vapour continuum absorption as some studies suggest. However, these continuum-induced uncertainties in 1D RCE computed clear-sky shortwave feedback are smaller than the uncertainties due to other sources such as the vertical distribution of RH (see, e.g., Bourdin et al., 2021). The uncertainties in shortwave feedback due to changes in continuum model obtained here are also much smaller than the overall spread of shortwave feedbacks across comprehensive climate models (e.g., Forster et al., 2021). But since the treatment of water vapour continuum is crucial for the correct computation of atmospheric radiative fluxes, the uncertainties in shortwave feedback from this work are non-negligible. More accurate laboratory and field measurements of the water vapour continuum in the

shortwave can constrain this source of feedback uncertainty and in turn contribute to reducing the discrepancies in the estimation of shortwave radiative feedback, especially in a warming world.

It has been hypothesised that thin clouds in the atmosphere create a longer pathlength for solar radiation and can thus enhance water vapour continuum absorption at transparency windows in the shortwave. In a future study, we plan to study the impact of this enhanced absorption on radiative feedback.

**Appendix A**

**Generation of correlated *k*-distribution gas-optics tables**

As stated in Section 1, the selected water vapour continuum models were parameterised in *k*-distribution gas-optics tables (or models) required for radiative transfer calculations in the RCE

model. For each water vapour continuum model, both the self- and foreign-continua were used in generating the gas-optics table.

        Recently, Hogan and Matricardi (2022) developed a flexible and efficient software tool (ECMWF correlated *k*-distribution, 'ecCKD') that can be used to generate accurate correlated *k*-distribution gas-optics models for radiation schemes of atmospheric models. These ecCKD

gas-optics tables use generally fewer *k*-terms (*g*-points) than other models, such as the widely used RRTM for GCM applications (RRTMG; Mlawer et al., 1997), and were shown to be very accurate under clear-sky conditions by validating them against line-by-line (LBL) radiative transfer calculations on independent data (Hogan and Matricardi, 2022).   The flexibility of the ecCKD tool allows the use of alternative water vapour continuum models for the training of *k*-

distribution gas-optics tables. This flexibility was exploited to generate the gas-optics tables used for this work.

        The Correlated K-Distribution Model Intercomparison Project (CKDMIP) datasets produced by Hogan and Matricardi (2020) were used to generate these gas-optics tables. Each of the CKDMIP datasets consist of spectral layer optical depths of 9 individual gases ($H_2O$, $O_3$,

$O_2$, $N_2$, $CO_2$, $CH_4$, $N_2O$, CFC-11 and CFC-12) in the shortwave spectral region from $250 - 50,000$ cm$^{-1}$. These optical depths were computed using the Line-By-Line Radiative Transfer Model (LBLRTM; Clough et al., 2005), version 12.8, which incorporates MT_CKD_3.2 water vapour continuum (and continua of other gases). The datasets used here are made up of 64

profiles for generating ecCKD gas-optics models and 50 profiles for independent evaluation

(see Table 3, Hogan and Matricardi (2020)).

To quantify the uncertainties in the water vapour continuum absorption on the gas-optics tables generated from these datasets, Hogan and Matricardi (2020) produced an additional set of water vapour profiles without the continuum. The continuum models selected for this work were added to these continuum-free profiles in turn and used together with the

profiles of the other gases to generate the gas-optics tables. Three ecCKD gas-optics tables, trained by each of the 3 water vapour continuum models described in Section 2.1, were generated as described in Hogan and Matricardi (2022) and Hogan (2022).

These gas-optics tables were generated for climate applications, as this is the focus of this work (see Table 1, Hogan and Matricardi, 2020); these tables enable calculations using a

large range of concentrations of the major anthropogenic greenhouse gases. The concentrations of gases and emission scenarios shown in Table 2 of Hogan and Matricardi (2020) were used. The "RGB" band structure was chosen for generating the gas-optics tables. In this band structure, the entire near-infrared spectral region is merged into one band, the visible region is divided into three bands for red, green and blue, and the ultraviolet region is merged into one

band. The heating-rate tolerance for each spectral interval ($g$-point) was set to 0.047 K d$^{-1}$. These specifications resulted in gas-optics models with 32 $k$-terms. See Hogan and Matricardi (2022) and Hogan (2022) for details of these specifications and other shortwave band structures that are available for use in ecCKD.

The generated gas-optics models were evaluated using LBL fluxes calculated from

them (as described in Hogan, 2022) and LBL fluxes from the 50 profiles independent dataset. Figures A1 shows the evaluation of the shortwave gas-optics model trained with MT_CKD_4.1.1 for present-day concentrations of the well-mixed greenhouse gases.

Figure A1 shows that the errors in the shortwave fluxes are about 1 W m$^{-2}$ or less at all vertical levels. This figure also shows that the heating rates errors are small, with the root-

mean-square error in the heating rates from the surface to the upper stratosphere being only about 0.057 K d$^{-1}$.

For the gas-optics tables trained with the other two water vapour continuum models (MT_CKD_2.5 and CAVIAR_updated), the errors in the shortwave upwelling fluxes, downwelling fluxes and heating rates are also small.

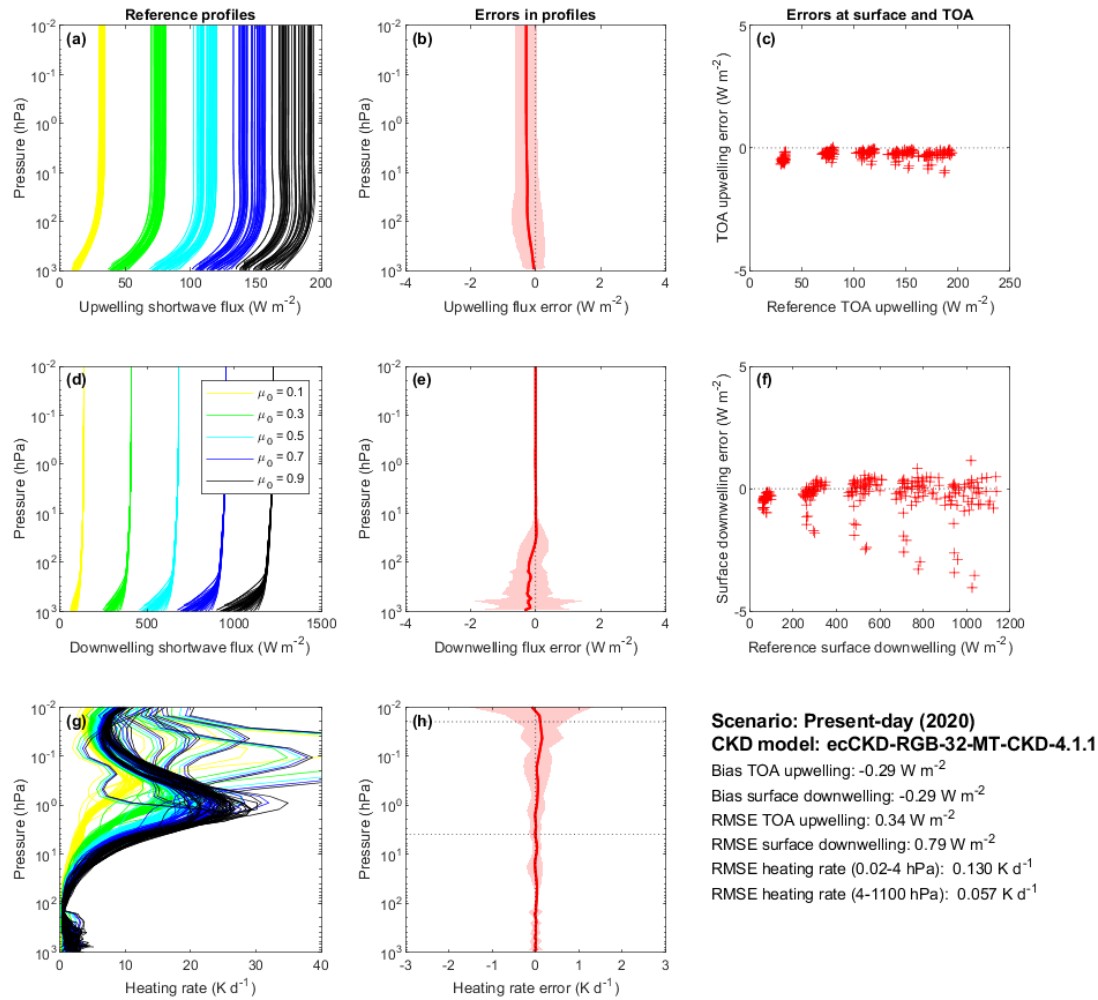

**Figure A1: An evaluation of clear-sky shortwave fluxes and heating rates from an ecCKD gas-optics table trained with the MT_CKD_4.1.1 continuum model for the 50 independent profiles of the CKDMIP evaluation data set with present-day concentrations of greenhouse gases. Upwelling fluxes, downwelling fluxes and heating rates from the**
535 **reference line-by-line calculations are shown in panels (a, d, g), while panels (b, e, h) show the corresponding biases in the calculations using the generated ecCKD gas-optics table. 95% of the errors are within the shaded areas. Panels (c and f) depict instantaneous errors in upwelling top-of-atmosphere and downwelling surface fluxes. The statistics of the comparison are summarized in the bottom right panel. The reference line-by-line**
**calculations in panels (a, d, g) are for all 50 CKDMIP evaluation profiles at five values of the cosine of the solar zenith angle, $\mu_0$ (0.1, 0.3, 0.5, 0.7, and 0.9). This gives a total of 250 combinations that are used in the subsequent evaluation.**

## Author contributions

KM and SB conceptualized and designed the experiments, and KM carried them out. LK and FR contributed to the development of the model code. RH and KM generated the gas-optics tables for the radiative transfer calculations. KM prepared the manuscript with input from all co-authors.

## Acknowledgments

This study contributes to the Cluster of Excellence "CLICCS — Climate, Climatic Change, and Society", and to the Center for Earth System Research and Sustainability (CEN) of the University of Hamburg. The authors thank Matthew Chandry and Oliver Lemke for their assistance in coupling ecRad to konrad. Manfred Brath is acknowledged for his assistance in simulating the terrestrial and solar spectrum shown in Figure 1. Kaah P. Menang was sponsored by the Alexander von Humboldt foundation under the program 'Humboldt Research Fellowship Programme for Experienced Researchers'. We also thank the anonymous referees for their insightful and constructive comments.

## Conflict of interest

The authors declare that they have no conflict of interest.

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
