# Peer review of "Variation in shortwave water vapour continuum and impact on clear-sky shortwave radiative feedback."

_EGUsphere, 2024_

## Author Comment (AC1)

Authors' response to reviewer comments

The authors thank the reviewer for his/her time and positive comments on their manuscript. These comments are addressed below. The necessary tracked changes, which hopefully will improve on the previous version, have been made to the paper. These comments are addressed on a point-by-point basis with his responses in red.

RC1
This manuscript evaluates how different shortwave water vapor continuum models impact the calculation of clear-sky shortwave radiative feedback, computed using ECMWF's ecRad radiative transfer code. The authors test three versions of the MT_CKD continuum model (v2.5, 3.2 and 4.1.1) as well as one version of the CAVIAR continuum model applied to ecRad. It is known that the strength of the SW water vapor continuum differs greatly across continuum models, but the impact of this on SW feedbacks has not been tested before (while such analysis has been done for the LW). Results presented here show that at a baseline temperature of 288K, the choice of MT-CKD continuum model version has a negligible effect on SW feedback. CAVIAR leads to a relatively stronger SW feedback than MT_CKD, but still only by a few percent. At moderately warmer baseline temperatures the differences in SW feedback across continuum models is larger. The manuscript is well written and fills a gap in the literature that will interest multiple groups of ACP readers. However, I think the importance of the results can be better motivated and there are some areas of the text where some minor clarifications would be helpful. I therefore recommend a minor revision.

– These positive comments and suggestion from the reviewer are highly appreciated.

General: My biggest concern with the manuscript is, to a reader coming from the GCM user community, the uncertainty in SW feedback associated with the different continuum models is quite small relative to other sources of uncertainty and to the overall spread in these feedbacks across GCMs, and certainly relative to overall net LW+SW feedback spread. Should the reader's takeaway be that continuum choice doesn't really matter, relatively speaking? Or is there a reason to care? The authors should spend some time addressing this in order to improve motivation of the work. In my mind, this work matters because the continuum is rooted in fundamental physics and observations. Therefore, in some respects, this is a source of feedback uncertainty that we can constrain. That is not true for many other sources of uncertainty.

– The authors thank the reviewer for these important questions and constructive discussion.
– We agree with the reviewer that the continuum induced uncertainty in shortwave feedback obtained in this work is small relative to other sources of uncertainty in 1D RCE computations (e.g,, vertical relative humidity profile). However, this uncertainty is non-negligible because the treatment of the water vapour continuum is of fundamental importance for the correct computation of atmospheric radiative fluxes (and by extension, radiative feedback). Thus, the choice of the water vapour continuum model matters.
– We also agree with reviewer that the uncertainties obtained in this work is relatively smaller than the overall spread of feedbacks across comprehensive climate models.
– As discussed in our manuscript (Section 1), there are still relatively significant uncertainties in characterising continuum absorption at some SW spectral regions. In

the concluding section of our manuscript, we pointed out that constraining water vapour continuum absorption in the shortwave will contribute to reduce the discrepancies in estimated shortwave feedbacks from 1D RCE models. We have now revised this section of the manuscript to put our results in a broader context as suggested by the reviewer.

Abstract: The result in Figure 4b and c, showing the varying dependency on surface temperature, and the author's spectral explanation of this result, is really interesting. I think some summary of this is worthy of inclusion in the abstract.  The result is much more nuanced than just "uncertainty is larger at warmer temperatures" as I assumed before reading this.

    – We thank the reviewer for this recommendation. The abstract has been revised accordingly.

Line 91-92: The authors should briefly summarize the results of the studies that investigated the effect of contiuum on LW feedback. It would help put this work into context.  My sense is LW feedback is similarly insensitive to continuum magnitude at present-day temps (288K).  Also, it would be helpful to understand how the range of continuum strength studied here compares to, for instance, the variations in continuum used by Koll et al. 2023.

    – The introductory section has been revised as recommended by the reviewer.

Figures 2 and 3 and surrounding text seem unnecessary and a bit out of place.  They are used to show that ecCKD is an accurate radiative transfer model, but that doesn't really have any baring on the main focus of the paper: the impact of continuum model on LW feedback.

    – Thank you for this comment. The main focus of this paper is the impact of shortwave continuum on SW feedback and not LW feedback as stated by the reviewer.
    – The figures and text the reviewer has referred to are in Sec 2.2 of the manuscript. The aim of this section is to help the reader understand how the correlated-$k$ tables for different continuum models were computed since the details of these computations have not been published elsewhere and there is no manuscript in preparation.  We think that without this section, it would be difficult to understand how various continuum models were implemented in ecRad.
    – Despite the importance of this section, we agree with the reviewer that it is a bit out of place at this portion of the manuscript. We have now moved it to an appendix. Additional we have limited our discussion to the shortwave only, as that is the focus of our paper.

Line 351: I'm not a fan of using the term "error" relative to 4.1.1, implying 4.1.1 is truth. The authors could use "difference" like fig 4 does

    – The text has been corrected as recommended.

Line 354-356: This qualitative explanation of why SW feedback is stronger at warmer temperatures isn't clear.  This same argument – increased moisture reduces upwelling radiation – could be said for lower temperatures.  A rewrite with a little more detail would be helpful here.
    – This portion of the manuscript has been re-written to clarify it.

---

## Author Comment (AC2)

Authors' response to reviewer comments

The authors thank the reviewer for his/her time and positive comments on their manuscript. These comments are addressed below. The necessary tracked changes, which hopefully will improve on the previous version, have been made to the paper. These comments are addressed on a point-by-point basis with his responses in red

RC2
The paper "Variation in shortwave water vapour continuum and impact on clear-sky shortwave radiative feedback" by Menang et al. provides an analysis of the differences in shortwave radiative feedback resulting from using different specifications of the water vapor continuum. This analysis uses idealized moist adiabat profiles, which is a standard approach used in similar studies, and is competently done. The results are clearly presented.
There are some serious issues with this manuscript that are detailed below. When they are resolved, it would still be questionable to me whether this study is of sufficient import to merit publication. The question it addresses doesn't seem to be one that people in the field are asking, and the result of the study indicates that the lack of interest may be due to people intuitively understanding that the shortwave water vapour continuum isn't too important for climate, as this study indicates (even if one accepts the premise of this paper that there is a fair amount of uncertainty in the shortwave continuum). I guess I would come down on the side of publishing this manuscript after revision, but the editors should know that it is a close call.
The main issue with this manuscript is that insufficient context is provided for the development of the different continuum versions, and the little context that is provided is misleading. The paper needs to provide more information about the observational foundation for each continuum version, enough so a reader can have an informed perspective on each version that is being analyzed.  An especially glaring omission is that no mention is made of the numerous recent laboratory studies by the Campargue group at Grenoble that provided the basis for the development of self continuum in MT_CKD_3.2 (and subsequent versions). These measurements were performed using the highly accurate cavity ring-down technique, and without that knowledge a reader would not have the context to evaluate the results presented in this manuscript.
The limited context on the continuum versions that the manuscript provides suggests that the CAVIAR provides a more up-to-date and accurate specification (e.g. lines 111-115 and 445-451) of the continuum, which is most likely the opposite of the situation. Prior to MT_CKD_3.2, it was clear from multiple observational studies that the self continuum in MT_CKD was too weak in near-infrared windows. Two sets of measurements were available at that time, those from CAVIAR and those from the Campargue group, which provided conflicting information about its strength. The MT_CKD developers decided that the cavity ring-down measurements, which are considered to be highly accurate, provided a better foundation for a new self continuum version than the CAVIAR measurements, which were performed with a technique that has difficulty measuring weak absorbers and provided results showing suspiciously flat behavior within and in consecutive near-infrared water vapor windows. The choice of using the Campargue group measurements was not a difficult one.

The later Elsey et al. study did nothing to modify this perspective – although its authors did a terrific job in a challenging analysis, the uncertainty in the specification of attenuation by aerosols made that result difficult to rely upon.

The authors of this manuscript must present context so readers can understand the basics of this situation and should not imply in any way that the CAVIAR continuum is considered to be more up-to-date since that would be misleading at best.

Another aspect of the context deficiencies in this manuscript is the lack of any mention whatsoever of the foreign continuum, as well as its role in the continuum versions utilized in this study. Even though it is less important than the self continuum in near-infrared windows, it would be remiss to not discuss it so that readers can understand its level of relevance to the subject at hand, especially since the Campargue group has also provided accurate laboratory measurements of its strength in these windows that have provided the foundation for a revision of MT_CKD.

Fig. 4 is an important to the result presented in this study, and its discussion (lines 398-406) is inadequate in several ways:

1. An intriguing thing about the results in Fig. 4 is that the shortwave feedback from CAVIAR is less than that due to MT_CKD at higher temperatures. One would think that this result is due to the absorption in CAVIAR being less than MT_CKD in some piece of the shortwave spectral region. However, no explanation is provided of why this happen -- previously the manuscript discusses only that CAVIAR is stronger than MT_CKD.

2. It is incorrect that the water vapor continuum has not been measured at > 8000 cm-1. The measurements in Campargue et al. 2016 go up to 8300 cm-1 and there is also a measurement by Fulghum and Tilleman near 10,000 cm-1. (See Fig. 13 of Campargue et al, 2016.)

3. It should be mentioned here that CAVIAR and MT_CKD are very close to each other at higher wavenumbers since (I believe) that CAVIAR was constructed to agree with MT_CKD above ~8000 cm-1.

4. The use of the word "extrapolated" here is likely to mislead readers. The authors should provide a basic explanation of how MT_CKD (and therefore CAVIAR, given the comment directly above) obtains its absorption coefficients in regions with no or limited measurements, i.e. through the derivation of a constrained line shape.

Additionally, in several instances the manuscript mentions the longwave region and the water vapour continuum in the longwave. These are not relevant to the subject of this paper and will confuse the reader more than they will help the reader. I recommend that all mentions of the longwave (including in figures) be removed, which should allow the distinction between versions 3.2 and 4.1.1 to be eliminated, thereby simplifying the study.

- These constructive comments and suggestions from the reviewer are highly appreciated.
- A brief description of the major revisions of the MT_CKD model relevant to our work has been included in the manuscript as suggested by the reviewer. Initially this was not included because we assumed that it has been adequately covered in the literature (although we did not point the readers to any). We thank the reviewer for pointing out this shortcoming.
- The authors have also revised and extended the discussion on CAVIAR continuum model. Of course, the authors do not consider this model to be more up-to-date than the MT_CKD model.

- Both the self and foreign continua are used in our work. Our discussion on the developments of the MT_CKD and CAVIAR models now includes both the self and foreign continua.
- We thank the reviewer for the comments related to Figure 4 of the submitted manuscript. The discussion relating to this figure has been revised to address the issues raised by the reviewer.
- We agree with the reviewer that including any discussion and analysis on the longwave continuum is irrelevant to the paper. The paper has been revised as recommended. This has of course led to the removal of MT_CKD_3.2 from our analysis.

Specific comments:

25 – MT_CKD versions are referred to with an underscore before the version number (e.g. MT_CKD_4.1.1) by its developers (e.g, see Mlawer et al., 2023), so this manuscript should follow that convention.

- Thank you for pointing this out. We have corrected the manuscript as recommended.

40 – no comma after "investigate".

- This typo has been corrected.

82 – "with little or no justification". The authors should explain what this means or remove it. By the context, it seems like the people that the authors believe have shown little or no justification for this selection are the respective developers of the climate models. I would think that these developers would have used the most recent version of the MT_CKD model at the time their RT codes were built. That seems very logical, but this text implies that this choice was made for no reason.

- We agree with the reviewer that the phrase 'with little or no justification' is misleading. Of course, climate model developers (and other users of the MT_CKD model) will use the most recent of the MT_CKD model, as it reflects the current understanding of continuum absorption. The misleading phrase has been removed from this sentence.

110-114 – Some mention should be made of the source of the CAVIAR continuum for high wavenumbers. As mentioned above, it agrees with MT_CKD_4.1.1, which I think it due to a choice made by the CAVIAR developers.

- More information on the CAVIAR continuum has been added to the manuscript.

110-122 – It would be useful to provide, in addition to the publications referenced here, the release years of each version.
Also, it would be worth notifying the readers that after this study was completed a significant change (v4.3) was made to the near-IR foreign continuum in MT_CKD that would impact the absorption of solar radiation. The authors could point the readers to https://github.com/AER-RC/MT_CKD_H2O/wiki/What's-New.
This development makes this study a little less up-to-date, but it should be mentioned for completeness.

- The release years for each continuum model has been included in the manuscript.
- We thank the reviewer for pointing us to the recent version of the MT_CKD model and for the suggestion to include it in the paper. This has been done.

117 – Since MT_CKD_3.2 is identical to MT_CKD_4.1.1, there is no reason to include both in the analysis since it just unnecessarily complicates the paper for a reader. Instead, just include v4.1.1 and mention that it includes a major revision to the self continuum compared to v2.5.

- Thank you for this comment. The text has been revised accordingly

166, 239-240 – The actual name of this code is RRTM for GCMs (not "Rapid Radiative Transfer Model…").

- The authors do not understand this correction from the reviewer. We understand that RRTM stands for 'Rapid Radiative Transfer Model' (see Mlawer et al 1997) while RRTMG stands for 'Rapid Radiative Transfer Model for GCMs' (see, e.g., http://rtweb.aer.com/rrtm_frame.html, Pincus et al 2019).

173-176 – This sentence is awkward, please rephrase.

- The sentence has been rephrased as recommended.

190 – I don't think it is sufficient to use the "RGB" jargon without further details provided in this manuscript. It shouldn't be up to the reader to go back to a previous paper to figure this out. Complete detail isn't necessary here, but enough so a reader understands the general principle behind this choice.

- This portion of the manuscript has been re-written with more details on 'RGB'.

190 – "resulted in"

- Corrected

200-202 – Is there a point to mentioning the longwave tables and showing the longwave results (203-205, Figure 2) in this paper? They seem irrelevant to the point of this paper.

- We agree with the reviewer that any results/discussion relating to the longwave in this section is irrelevant to this paper and have now been removed.
- We have also moved this section of the manuscript to an appendix because it has no direct bearing on the main focus of the paper.

240-241 – It is unclear why Pincus et al. is used as a reference for how widely used RRTMG is.

- Appropriate references for how widely used RRTMG is have been provided.

255 – "solar constant" is well-established term in science and has a value of ~1360 W/m2. Another term should be used here for the quantity (e.g. extraterrestrial solar irradiance"?) being referred to.

  – Thank you for this comment. We have replaced 'solar constant' with 'reduced total solar irradiance'

256 – "with no diurnal cycle" - It seems clear from the details provided in this section that no diurnal cycle is being used, so perhaps it is more confusing than elucidating to include this phrase.

  – The phrase 'with no diurnal cycle' has been removed from the manuscript.

261 – "an albedo of 0.2."

  – This has been corrected

262, 269, 272, 278 – "was".

  – Corrected

288 - "adjusted"

  – Corrected

289 – "calculated"

  – Corrected

The present tense is being used elsewhere in this section.

Section 3.1.1 – This section is completely unnecessary since it doesn't deal with the shortwave nor the water vapor continuum. Therefore, having it be the first result in the "Results" section is confusing. It should be removed from the paper. The statement at line 321-322 can be modified to directly refer to the (say) Kluft et al. value of -1.8 W m-2 K-1.

  – Section 3.1.1 has been removed from the manuscript as recommended by the reviewer. Line 321-322 has also been revised.

324-330 – These sentences are repetitive and should be streamlined.

  – Thank you for this comment. The sentences have been streamlined as recommended.

337 – No comma after "expected".

  – Corrected

348 – No comma after "temperature".

– Corrected

351, 367, 371 - "difference" would be better than "error" and consistent with the axis labels in Fig. 4.

– Thank you for this comment. The manuscript has been revised as recommended.

371 – Perhaps "magnitude of the relative error" since it is negative.

– We agree with reviewer. The manuscript has been revised.

386-387 – "Shortwave water vapour absorption in windows between absorption band…". Without such a qualification, the current sentence is incorrect.

– The authors appreciate this correction. The sentence has been corrected.

389 – "atmospheric"

– Corrected

398 – Would "contributing to the substantial differences in lambda_SW there" be closer to the authors' meaning than the current text? (similar comment related to the use of the word "uncertainty" on line 400.)

– The authors are referring to the results presented in their work and thus 'differences' would be a more appropriate word as suggested by the reviewer.

423 – Perhaps "a range of fixed surface temperatures…"

– Thank you for this suggestion. The manuscript has been revised accordingly.

430 – The use of a semi-colon here is awkward.

– The semi-colon has been and this portion of the manuscript re-written.

434, 439 – As mentioned above, "error" is not a good way to refer to this since there is no reference result with respect to which an error can be defined.

– We agree with the reviewer. This has been corrected.

---

## Author Response (AR2)

Authors' response to reviewer comments

We thank the reviewer for taking his/her time to review our manuscript again. As was the case during the first review, the reviewer's comments are addressed on a point-by-point basis with his responses in red. The necessary tracked changes have been made to the manuscript.

The paper "Variation in shortwave water vapour continuum and impact on clear-sky shortwave radiative feedback" by Menang et al. has improved since the initial submission, with all major problems resolved. I commend the authors for the efforts to improve the manuscript. With a few more minor modifications, this paper should be published.

Thank you very much for your positive comments

I do recommend a small amount of further modification in the continuum background section (2.1). Although the continuum version evolution presented in the main paragraph has all the relevant facts, I found the many shifts in the narrative between the self and the foreign continuum to be confusing. Instead, I suggest that the authors present the self continuum information first, and then present the foreign continuum information in a second paragraph. This would allow both clearer exposition and provide an opportunity for the authors to further emphasize that the self continuum is more important than the foreign for what they are considering in this study.

Also, I found the statement about Elsey et al. on line 155 out of place. The paragraph is focused on the evolution of MT_CKD and its close relationship with the various laboratory studies by the Campargue group, and this sentence is not about that. Also, that sentence points out that Elsey et al. indicate "MT_CKD_3.2 is also underestimating the self-continuum", while the previous sentence indicates that MT_CKD_3.2 is stronger than Vasilchenko et al. Therefore, the word "also" seems to not make sense.

Perhaps the authors do not agree with the perspective in my previous review that the experimental techniques used by the Campargue group are superior to the Fourier transform spectrometer approach. Even if that is the case, I think that the authors could still provide their readers with additional context by stating that the Campargue group techniques are highly regarded, which I hope the authors believe. This could be done by inserting text in the sentence that begins on line 144, such as "Optical-feedback-cavity enhanced absorption spectroscopic and cavity ring-down spectroscopic laboratory measurements, considered to be highly accurate, …"

- ➢ We thank the reviewer for these constructive comments and suggestions.
- ➢ We have presented the self and foreign continua separately as recommended by the reviewer.
- ➢ We agree with the reviewer that the OFCEAS and CRDS measurements are more accurate than FTIR measurements. We apologise that we did not highlight that in our revised manuscript. We have now inserted the text proposed by the reviewer in the paper. We thank the reviewer for this.
- ➢ Elsey et al. was included in this paragraph as one of the studies that validated MT_CKD_3.2. Unfortunately, this instead made the paragraph to be confusing. With

the revision of this portion of the manuscript as recommended by the reviewer, Elsey et al. has been removed.

> We have also emphasized the importance of self-continuum for our work as recommended by the reviewer.

227-228 – The authors are incorrect about the name of RRTMG. It is simply "RRTM for GCM applications". See section 2 in DOI: 10.1175/AMSMONOGRAPHS-D-15-0041.1 This needs to be corrected before the paper is published.

We thank the reviewer for pointing us to formal name of RRTMG. This has now been corrected.

285 – Add space after "2.5".

Done

337 – "temperatures"

Corrected

416 – "laboratory-implied" is awkward (and maybe not a word). Maybe "observation-based" would be better? Even with this change, this sentence is little misleading in that in the spectral region that is the focus of this paper both MT_CKD_4.1.1 and CAVIAR_updated are constrained to laboratory measurements.

We thank the reviewer for this comment. We agree with the reviewer that the word 'laboratory-implied' makes this sentence to be misleading. Since the origin of the MT_CKD and CAVIAR_updated models have already been discussed in the manuscript, we have removed both 'semi-empirical' and 'laboratory-implied' from this sentence to avoid any misleading information.

421-428 - There are several past tense verbs in this paragraph that probably should be present tense.

The past tense verbs have been corrected as pointed out by the reviewer.